# Prolyl Hydroxylase Domain-Containing Protein 3 Gene Expression in Chondrocytes Is Not Essential for Bone Development in Mice

**DOI:** 10.3390/cells10092200

**Published:** 2021-08-26

**Authors:** Weirong Xing, Sheila Pourteymoor, Gustavo A. Gomez, Yian Chen, Subburaman Mohan

**Affiliations:** 1Musculoskeletal Disease Center, Jerry L. Pettis VA Medical Center, Loma Linda, CA 92357, USA; weirong.xing@va.gov (W.X.); Sheila.Pourteymoor@va.gov (S.P.); Gustavo.Gomez2@va.gov (G.A.G.); yian.chen@va.gov (Y.C.); 2Departments of Medicine, Loma Linda University, Loma Linda, CA 92354, USA; 3Departments of Biochemistry, Loma Linda University, Loma Linda, CA 92354, USA; 4Departments of Physiology, Loma Linda University, Loma Linda, CA 92354, USA

**Keywords:** prolyl hydroxylase, bone development, chondrocyte, bone, gene knockout

## Abstract

We previously showed that conditional disruption of the *Phd2* gene in chondrocytes led to a massive increase in long bone trabecular bone mass. Loss of *Phd2* gene expression or inhibition of PHD2 activity by a specific inhibitor resulted in a several-fold compensatory increase in *Phd3* expression in chondrocytes. To determine if expression of PHD3 plays a role in endochondral bone formation, we conditionally disrupted the *Phd3* gene in chondrocytes by crossing *Phd3* floxed (*Phd3^flox/flox^*) mice with *Col2α1-Cre* mice. Loss of *Phd3* expression in the chondrocytes of *Cre^+^*; *Phd3^flox/flox^* conditional knockout (cKO) mice was confirmed by real time PCR. At 16 weeks of age, neither body weight nor body length was significantly different in the *Phd3* cKO mice compared to *Cre^−^*; *Phd3^flox/flox^* wild-type (WT) mice. Areal BMD measurements of total body as well as femur, tibia, and lumbar skeletal sites were not significantly different between the cKO and WT mice at 16 weeks of age. Micro-CT measurements revealed significant gender differences in the trabecular bone volume adjusted for tissue volume at the secondary spongiosa of the femur and the tibia for both genotypes, but no genotype difference was found for any of the trabecular bone measurements of either the femur or the tibia. Trabecular bone volume of distal femur epiphysis was not different between cKO and WT mice. Histology analyses revealed *Phd3* cKO mice exhibited a comparable chondrocyte differentiation and proliferation, as evidenced by no changes in cartilage thickness and area in the cKO mice as compared to WT littermates. Consistent with the in vivo data, lentiviral shRNA-mediated knockdown of *Phd3* expression in chondrocytes did not affect the expression of markers of chondrocyte differentiation (*Col2*, *Col10*, *Acan*, *Sox9*). Our study found that *Phd2* but not *Phd3* expressed in chondrocytes regulates endochondral bone formation, and the compensatory increase in *Phd3* expression in the chondrocytes of *Phd2* cKO mice is not the cause for increased trabecular bone mass in *Phd2* cKO mice.

## 1. Introduction

In mammals, all three prolyl hydroxylase domain (PHD) enzymes including PHD1, PHD2 and PHD3 share a highly conserved hydroxylase domain in the catalytic C-terminal regions, whereas the N-terminal regions are more divergent and with no known functions [1]. Both PHD1 and PHD2 contain more than 400 amino acid residues, while PHD3 has less than 250 with a short N-terminal sequence. Recent studies have found that PHDs are negative regulators of the hypoxia-inducible transcription factor (HIF)1α and HIF2α [2,3]. The hydroxylation of specific proline residues (Pro-402 and Pro-564) in the oxygen-dependent degradation domains (ODDs) of the HIF1α by PHDs leads to the targeting of HIF1α for ubiquitination through an E3 ligase complex initiated by the binding of the Von Hipple Lindau protein (pVHL) and subsequent proteasomal degradation [2,3]. It has been known that PHD1 and PHD2 are able to hydroxylates the C-terminal ODD (CODD) but are more active for the N-terminal ODD (NODD) whereas PHD3 almost exclusively hydroxylates the CODD [4,5]. Hydroxylation of HIF1α requires molecular oxygen and iron. Under the hypoxia condition, PHDs are inhibited, and the HIF1α accumulates in the cytoplasm, translocates to the nucleus, interacts with the p300/CBP co-activator. The complex protein then binds to DNA to regulate hypoxia-responsive genes including *Vegf*, *Runx2* and *Osterix* (*Osx*) [6,7]. Besides HIF1α involvement in the regulation of angiogenesis and osteogenesis during skeletal development, other reports demonstrated that PHD can hydroxylate other substrates including IKK-β, β2-adrenergic receptor, HIF1α binding protein-suppressor of cytokine signaling (SOCS), and Argonaute (Ago) and can influence their functions in a number of ways [8,9,10,11]. Mice with disruption of *Phd1* and *Phd3* genes survive normally, but *Phd2* knockout (KO) embryos die at mid-gestation stages due to the defects of underdeveloped placenta [12].

PHDs play important roles in bone development and homeostasis. Of the three *Phd* family members, *Phd2* is highly expressed in bone and targeted KO of the *Phd2* gene in osteoblasts using *Col1α2-Cre* results in reduced bone mass in the trabeculae of long bones by downregulating *Osx* expression [13]. By contrast, PHD2 serves as a negative regulator for endochondral bone formation, as the chondrocyte-specific KO mice displayed a dramatic increase of bone mass in the trabeculae of long bones and spines caused by increased HIF signaling in chondrocytes [14]. The trabecular number and thickness were significantly increased, but trabecular separation was reduced in the *Col2α1-Cre* directed conditional KO (cKO) mice. Cortical thickness and tissue mineral density at the femoral mid-diaphysis of the cKO mice were also significantly increased. In our studies on the mechanism by which PHD2 regulates chondrocyte differentiation and increased bone formation, we found that loss of PHD2 function in chondrocytes resulted in marked upregulation of *Phd3* expression, both *in vitro* and *in vivo*. While the expression of *Phd1* was not changed, the expression of *Phd3* was significantly increased by two-fold in the growth plates of cKO mice. Knockdown of *Phd2* expression in primary chondrocytes caused a seven-fold induction of *Phd3* expression [14]. To determine if PHD2 effects on trabecular bone formation is in part caused by elevated *Phd3* expression, we evaluated the consequence of disruption of the *Phd3* gene in chondrocytes on peak bone mass in mice.

## 2. Materials and Methods

### 2.1. Generation of Conditional Knockout Mice

*Phd3* floxed mice kindly provided by Dr. Guo-Hua Fong (University of Connecticut School of Medicine, Fermington, CT 06032, ME, USA) were first crossed with the *Col2α1**-Cre* transgenic line to generate *Cre^+^; Phd3* loxP heterozygous mice (*Phd3^flox/+^; Col2α1-Cre^+^*) [15,16,17]. The *Phd3^flox/+^; Col2α1-Cre^+^* mice were then backcrossed with *Phd3^flox/flox^* mice to generate *Phd3**^flox/flox^*; *Col2α1-Cre^+^* cKO mice and *Cre^−^*; *Phd3^flox/flox^* or *Phd3*^flox/+^ wild-type (WT) littermates (Figure 1A). The genetic background of these mice is C57BL/6. Both genders were used in the studies. Animals were housed at the Jerry L. Pettis Memorial VA Medical Center (Loma Linda, CA, USA) according to approved standards with controlled temperature (22 °C) and illumination (14 h light, 10 h dark), as well as unlimited food and water. Animal procedures were approved by the Institutional Animal Care and Use Committee of the Jerry L. Pettis Memorial Veterans Affairs Medical Center. Mice were anesthetized with isoflurane prior to the procedures. The animals were euthanized by exposure to carbon dioxide followed by cervical dislocation.

### 2.2. Antibodies and Biological Reagents

Polyclonal anti-collagen 10 antibody (Cat No: ab58632) was purchased from Abcam (Abcam, MI). MISSION^®^ shRNA Lentiviral Transduction Particles against *Phd3* and control non-target were purchased from Sigma. The hairpin sequences of targeting shRNA were below

Mouse *Phd3*-TRCN0000009753:CCGGCGGCTTCTGCTACCTGGACAActcgagTTGTCCAGGTAGCAGAAGCCGTTTTTNon-target:CCGGCAACAAGATGAAGAGCACCAActcgagTTGGTGCTCTTCATCTTGTTGTTTTT

### 2.3. Evaluation of Bone Phenotypes

Areal bone mineral density (aBMD) and bone area of the total body as well as femur, tibia, and lumbar vertebra (L4–6) of 16-week old mice were measured by the FAXITRON UltraFocus^DXA^ 1000 In Vivo Imaging and DXA Analysis System (FAXITRON Bioptics, LLC, Tucson AZ 85706, USA) under anesthetization according to the manufacturer’s instruction. Trabecular and cortical bones of the femur, and L5 vertebra were analyzed by microcomputed tomography (µCT; VIVA CT40, SCANO Medical) in 16-week old mice as reported previously [18]. The µCT scanning was performed with 55–70 kVp volts (55 kVp for trabecular bone, 70 kVp for cortical bone) and a voxel size of 10.5 micron, and microarchitecture reconstructions were performed using the SCANCO software (SCANO Medical). Multiple sections of 1.05 mm cortical bone in the femoral mid diaphysis were used to analyze long-bone cortical parameters, and the average measurements of the multiple sections were used for data presentation. A 1.89 mm length (180 sections) of the secondary spongiosa of the femurs starting at 0.3675 mm (35 sections) proximal to the growth plate were analyzed for trabecular bone parameters. For the cortical bone, 20 slices were analyzed at the femoral and tibia midshafts. The exact numbers and location of slices used for analyses were adjusted for length so that the analyzed regions were anatomically comparable between samples. To analyze epiphysis formation, the distal femoral epiphyses were used to evaluate the total tissue volume (TV mm^3^), bone volume (BV, mm^3^), bone volume fraction (BV/TV, %) as described previously [19,20,21].

### 2.4. Histology and Immunohistochemistry

Long bones from 16-week old mice were fixed in 10% formalin overnight, washed, dehydrated and embedded in paraffin for sectioning. Cartilage of the distal femur was stained with safranin O and counter-stained with hematoxylin. Articular cartilage width and area of the distal femurs were measured in a blinded fashion with computer software OsteoMeasure (OsteoMetrics, Decatur, GA, USA) [22,23]. Immunohistochemistry was performed using a rabbit immunohistochemistry kit (Vector Laboratories, Burlingame, CA, USA). Briefly, femoral epiphyseal sections were de-paraffinized in HistoChoice clearing agent, rehydrated in a graded series of ethanol and tap water, and treated with 3% H_2_O_2_ for 30 min to inactivate endogenous peroxidase activity. The sections were then rinsed with PBS (pH 7.4) and heated for 20 min at 90 °C in sodium citrate citric acid buffer (pH 2.5) for epitope recovery. The sections were pretreated with a blocking solution containing normal goat serum for 20 min, and then incubated with anti-collagen 10 antibody at a dilution of 1:100. Positive and negative control sections were incubated with anti-β-actin (Sigma) and normal rabbit IgG, respectively. After an overnight incubation at 4 °C, the sections were rinsed with PBS, and incubated with biotinylated secondary antibodies for 30 min at room temperature. The slides were then washed in PBS, incubated with the VECTASTAIN ABC-AP kit (Vector Laboratories) for 30 min, rinsed again with PBS, and incubated with the Vector Red substrate until the desired color stain developed.

### 2.5. Cell Culture

Primary chondrocytes were isolated from the rib cartilage of 2-week old C57BL/6 mice (two pairs of female and male mice) and cultured as previously described [24]. Cells were grown in α-MEM medium containing 10% fetal bovine serum (FBS), penicillin (100 U/mL), and streptomycin (100 μg/mL) to approximately 30% confluence and transduced with shRNA- lentivirus particles at a multiplicity of infection of 10 in the presence of 8 µg/mL of polybrene. The medium was changed 12 h after infection, and the cells were grown for an additional 48 h for RNA extraction.

### 2.6. RNA Extraction and Real-Time Quantitative Polymerase Chain Reaction

Total RNA was extracted from chondrocytes or distal femur epiphysis with the Trizol reagent as described previously [25,26]. An aliquot of RNA (2 µg) was reverse-transcribed into cDNA in 20 µL volume of reaction by oligo(dT)_12–18_ primer. A real time PCR contained 0.5 µL template cDNA, 1x SYBR GREEN master mix (Qiagen), and 100 nM of specific forward and reverse primers in a 25 μL volume of reaction. Primers used for real-time PCR are listed in Table 1. Relative gene expression was determined by ^ΔΔ^CT method [27].

### 2.7. Statistical Analysis

Data were analyzed by Student’s *t*-test or ANOVA as appropriate. Two-sided tests were performed using STATISTICA software (Statsoft, Tulsa, OK, USA). Data are presented as Mean ± SEM (*n* = 6 per genotype for each gender).

## 3. Results

### 3.1. Conditional Knockout of Phd3 in *Col2α1*-Expressing Cells Does Not Impair Skeletal Development in Mice

To disrupt the expression of *Phd3* in chondrocytes, we generated *Phd3* cKO mice by crossing the *Phd3**^flox/flox^* mice with the *Col2α1**-Cre* transgenic mice, in which the Cre was found to be specifically expressed in *Col2α1*-expressing cells [14,17]. After two generations of breeding, the *Phd3**^flox/flox^*; *Col2α1*-Cre^+^ cKO mice were generated and compared to *Phd3**^flox/flox^* or *Phd3*^flox/+^; Cre^−^ WT littermates. The cKO mice were born alive with the expected Mendelian frequency and grew normally. To test whether *Phd3* is disrupted in the bones of cKO mice, total RNA was extracted from the distal femur epiphysis and growth plate region of 16-week old cKO and WT mice and used for real-time PCR with specific primers. Figure 1B shows that the *Phd3* mRNA level was reduced by 59% in the cKO mice as compared to the WT control mice. The expression level of *Phd2* was not affected in the epiphyses of *Phd3* cKO mice. At 16 weeks of age, neither body weight nor body length were significantly different in the cKO mice compared to gender-matched control mice for either females or males (Figure 2A).

Our DXA analyses found that total body, femur, tibia and lumbar BMDs were unchanged in both female and male cKO mice compared to gender-matched control mice (Figure 2B,C). While the BMD of total body, femur, and tibia of male cKO mice were slightly less compared to WT mice, these changes were not statistically significant (*p* > 0.05). Consistent with DXA data, μCT analyses of the trabecular bone of the femurs from 16-week old mice revealed that neither BV/TV nor any of the trabecular bone parameters including connectivity density (Conn-Dens.), trabecular number (Tb. N), trabecular thickness (Tb. Th) and trabecular spacing (Tb. Sp) in *Phd3* cKO mice were significantly different from the gender-matched WT mice for either genders (Figure 3A,B). Disruption of the *Phd3* gene in chondrocytes did not affect the cortical BMD or BV/TV of the femurs either (Figure 4A,B). The tibial BV/TV, BMD, Conn-Dens., Tb. N, Tb. Th, and Tb. Sp in cKO mice were comparable to those from WT mice for either genders (Figure 5A). Deficiency of *Phd3* expression in chondrocytes had no effect on cortical BV/TV and BMD of the tibia either (Figure 5B).

### 3.2. Disruption of Phd3 Expression in Col2α1-Expressing Cells Does Not Affect Chondrocyte Differentiation and Epiphysis Development in Mice

To investigate whether disruption of the *Phd3* gene in *Col2α1* expressing cells affects chondrocyte differentiation and epiphysis development and growth in mice, we examined the distal femoral growth plates, which contain mostly chondrocytes that contribute to endochondral bone formation, from the cKO and WT control mice. Our histological analyses found that chondrocyte differentiation was not affected in cKO mice as compared to WT controls. There was no change in the expression level of collagen 10 in the growth plate of cKO mice, as evidenced by immunohistochemistry. The decalcified knee joint sections showed that there were no changes in the width and area of the articular cartilage in the cKO mice as compared to WT mice for both genders (Figure 6A–C). Micro-CT analyses found that there was no difference of BV/TV of the femoral epiphyses between the cKO and WT mice (Figure 6D). To further confirm that disruption of *Phd3* expression in chondrocyte has no effect on chondrocyte differentiation, we isolated primary chondrocytes from the ribs of two pairs of two-week old female and male mice, and knocked down the expression of *Phd3* in the primary chondrocyte cultures by infecting the cells with lentivirus shRNA against to mouse *Phd3* gene or Lentivirus scramble control shRNA. *Phd3* mRNA was reduced by 34% (*p* < 0.01) in the lentivirus-*Phd3* shRNA transduced cells (Figure 6E). Expression levels of chondrogenic markers, *Col2α1* and *aggrecan* were not changed between the knockdown cells expressing shRNA against *Phd3* and the control chondrocytes expressing scramble shRNA. Expression of chondrocyte differentiation marker *Col10* was unchanged in the knockdown chondrocytes. Expression level of transcription factor *Sox9* was comparable in the *Phd3* knockdown cells as compared to control chondrocytes (Figure 6E).

## 4. Discussion

Three *Phds* are widely expressed in different organs at the transcript level. *Phd1* is expressed at the highest level in testes, whereas *Phd2* is the highest in the heart [28]. At the protein level, however, PHD2 is the most abundant in all mouse organs including bones [29]. In our previous studies, we found that PHD2 was predominantly expressed in osteoblasts and played an important role in regulating bone formation by modulating expression of Osx and bone formation marker genes [13]. Targeted disruption of *Phd2* in osteoblasts leads to short stature and premature death at 12 to 14 weeks of age. BMD in femurs and bone volume fraction (BV/TV) in the femoral trabecular bones of cKO mice were significantly decreased. In contrast, mice with conditional disruption of *Phd2* in chondrocytes were born normal, but quickly became growth-retarded due to increased cartilage matrix mineralization. The *Phd2* cKO mice exhibited increased endochondral bone formation in multiple skeletal sites including long bones and vertebrae [30]. Interestingly, the expression levels of *Phd3* were markedly elevated in both growth plate chondrocytes of the chondrocyte-specific *Phd2*-cKO mice *in vivo* and in Ad-Cre-mediated knockdown of *Phd2* expression primary chondrocytes *in vitro*. Our observation raised a question as to whether the compensatory increase in PHD3 expression in chondrocytes lacking PHD2 contributes to increased endochondral bone formation observed in the chondrocyte-specific *Phd2 c*KO mice. To test this possibility, we conditionally disrupted the *Phd3* gene in the Col2α1 expressing cells by crossing *Phd3* floxed mice with *Col2α1-Cre* mice. We found that neither body weight nor body length was significantly different in the *Phd3* cKO mice compared to *Cre^−^*; *Phd3^flox/flox^* WT mice at 16 weeks of age. Areal BMDs of total body, femur, tibia, and lumbar skeletal sites were not significantly different between the cKO and WT mice either. Micro-CT measurements revealed significant gender differences in the trabecular bone volume adjusted for tissue volume at the secondary spongiosa of the femur and tibia for both genotypes, but no genotype difference was found for any of the trabecular bone measurements of either femur or tibia. There was no change in the bone volume of the epiphysis growth plate of the femur in the cKO mice as compared to the gender-matched WT littermates. In agreement with the µCT data, our histology analyses did not reveal changes in articular cartilage thickness and area. The chondrocytes differentiated normally in the knee joints of the cKO mice. Consistent with the *in vivo* data, lentiviral shRNA-mediated partial knockdown of *Phd3* expression in chondrocytes did not affect expression of markers of chondrocyte differentiation (*Col2*, *Col10*, *Acan*, *Sox9*). These data as well as the findings from our previous study suggest that *Phd2* but not *Phd3* expressed in chondrocytes regulates endochondral bone formation, and the compensatory increase in *Phd3* expression in the chondrocytes of *Phd2* cKO mice is not the cause for increased trabecular bone mass in *Phd2* cKO mice [14]. In future, we will confirm the role of PHD2 deficiency induced PHD3 overexpression in chondrocyte differentiation and growth plate development in *Phd2/3* double knockout mice. We will also verify the expression levels of the chondrocyte differentiation markers in primary chondrocytes derived from the *Phd3* global KO mice.

PHD enzymes catalyze hydroxylation of the specific proline and asparagine residues of their substrates, such as hypoxia-inducible factor (HIF)-α [31]. Both HIF1α and HIF2α contain two sites of prolyl hydroxylation sites (e.g., the NODD and CODD). Prolyl-4 hydroxylation at two sites within a central degradation domain of HIF1α by PHDs, mainly PHD-2, mediates interactions with the VHL E3 ubiquitin ligase complex that targets HIF1α for proteasomal degradation [1,32]. Hydroxylation of an asparaginyl residue in the C-terminal activation domain by PHDs inhibits transcriptional activity by preventing interaction with co-activator, p300/CBP and inactivates HIF1α functions [33]. Recent studies have suggested that there exits PHD substrate specificity and preference. PHD3 was found to hydroxylate CODD but not NODD sequences, while PHD1 and PHD2 hydroxylate both CODD and NODD. PHD1 and PHD3 are more active on HIF2α than HIF1α, whereas PHD2 was more effective on the HIF1α [5]. In agreement with this study, mice with disruption of *Phd2* had an accumulation of HIF1α but not HIF2α in the liver and kidneys, whereas mice with deficiency of both PHD1 and PHD3 had hepatic accumulation of HIF2α but not HIF1α [29,34]. On the other hand, conditional disruption of *Hif1α* in cells of osteoblastic lineage impaired skeletal development [35,36]. These studies indicate that the differential effects on HIF1α versus HIF2α is determined by NODD rather than CODD sequences and hydroxylation. Furthermore, mice with conditional disruption of *Hif1α* in the condensing mesenchyme had shortened bones, less-mineralized skulls and widened sutures due to massive apoptosis and altered proliferation of chondrocytes in the growth plate [35]. By contrast, mice lacking *Hif-2α* had only a modest decrease in trabecular bone volume [37]. Because PHD2 is found to be localized in the nucleus, and it influences hydroxylation of methyl cytosine of HIF1a and VEGF promoters, it is possible that PHD2 and PHD3 may epigenetically regulate different promoters of transcription factors that are essential for chondrocyte differentiation [38]. Accordingly, our studies together with others provide evidence that PHD3 is not essential for regulation of HIF1α stability and skeletal development in mice.

## Figures and Tables

**Figure 1 cells-10-02200-f001:**
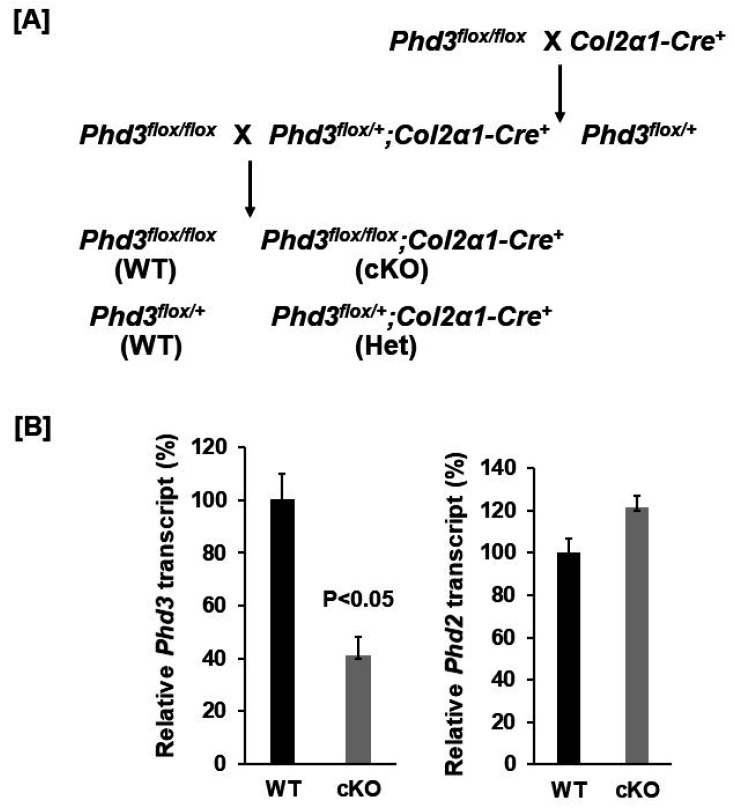
Generation of chondrocyte specific *Phd3* KO mice. (**A**): A breeding strategy of generation of chondrocyte specific *Phd3* cKO mice and control WT mice. (**B**): *Phd3* expression was partially disrupted in the bones of cKO mice. Total RNA was extracted from the distal femur epiphysis and growth plate region of 16-week old cKO and WT female mice and used for real-time PCR (*n* = 3).

**Figure 2 cells-10-02200-f002:**
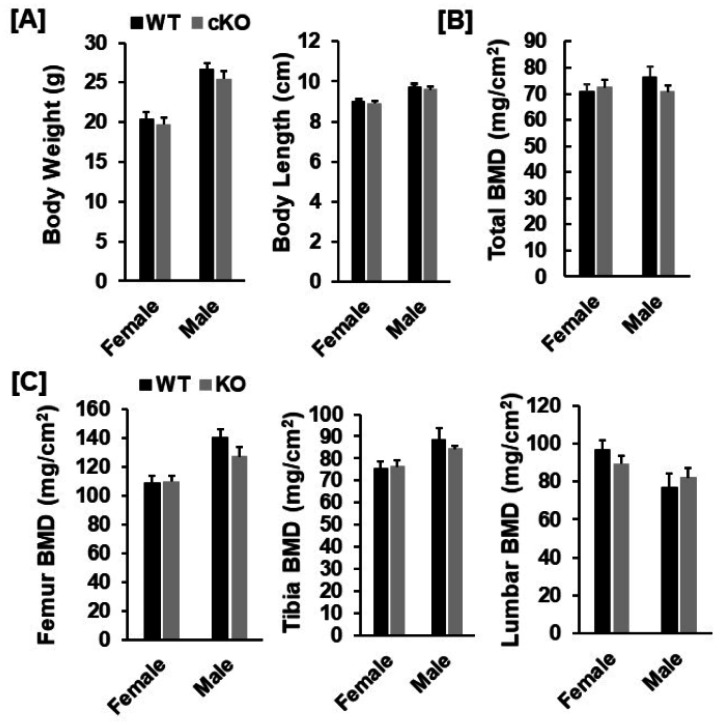
DXA analyses revealed no significant changes in bone parameters in the *Phd3* cKO mice at 16 weeks of age. (**A**): Body weight and body length of the WT and cKO mice, respectively. (**B**,**C**): Total body, femur, tibia, and lumbar BMD of the WT and the cKO mice, respectively, measured by DXA.

**Figure 3 cells-10-02200-f003:**
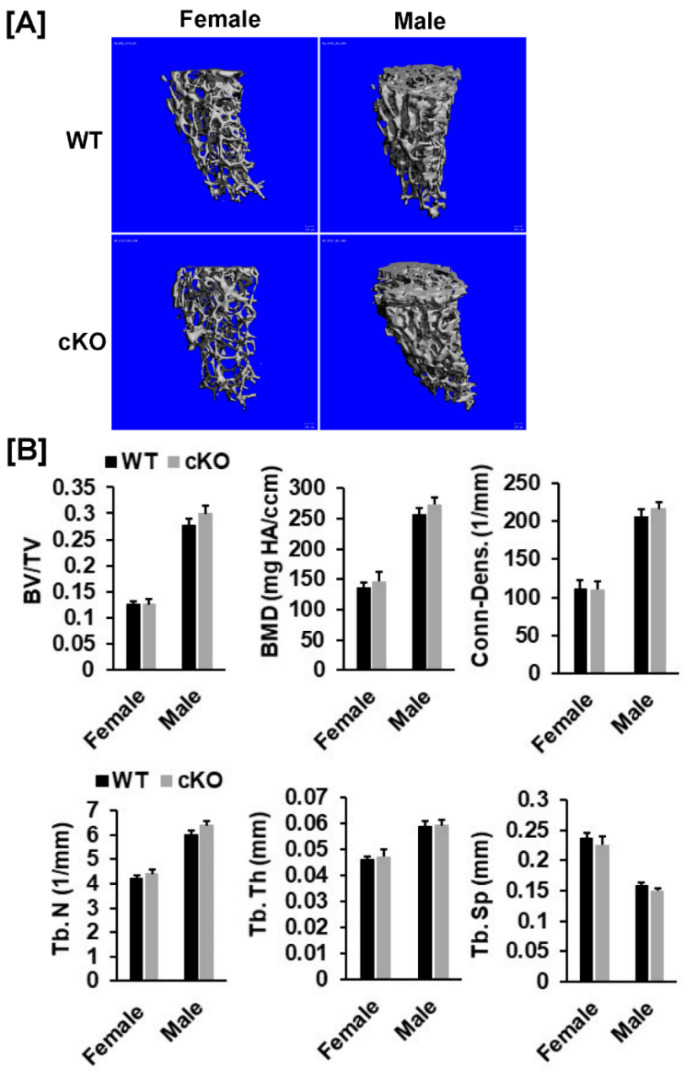
Micro-CT analysis revealed no changes in trabecular parameters of the femur of the *Phd3* cKO mice at 16 weeks of age. (**A**): Micro-CT images of the trabecular bone of the distal. (**B**): Quantitative data of trabecular parameters of the femur (BV/TV, BMD, Conn-Dens., Tb. N, Tb. Th, and Tb. Sp).

**Figure 4 cells-10-02200-f004:**
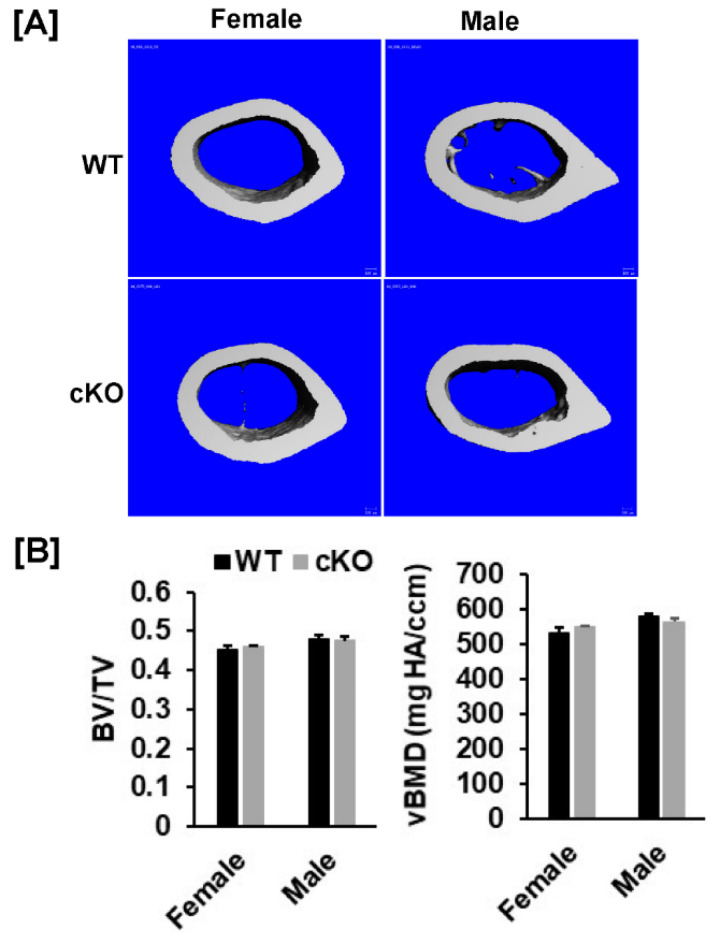
There were no changes in cortical parameters of the femur of the *Phd3* cKO mice at 16 weeks of age. (**A**): Micro-CT images of the cortical bone of the femur. (**B**): Quantitative data of cortical BV/TV and BMD of the femur.

**Figure 5 cells-10-02200-f005:**
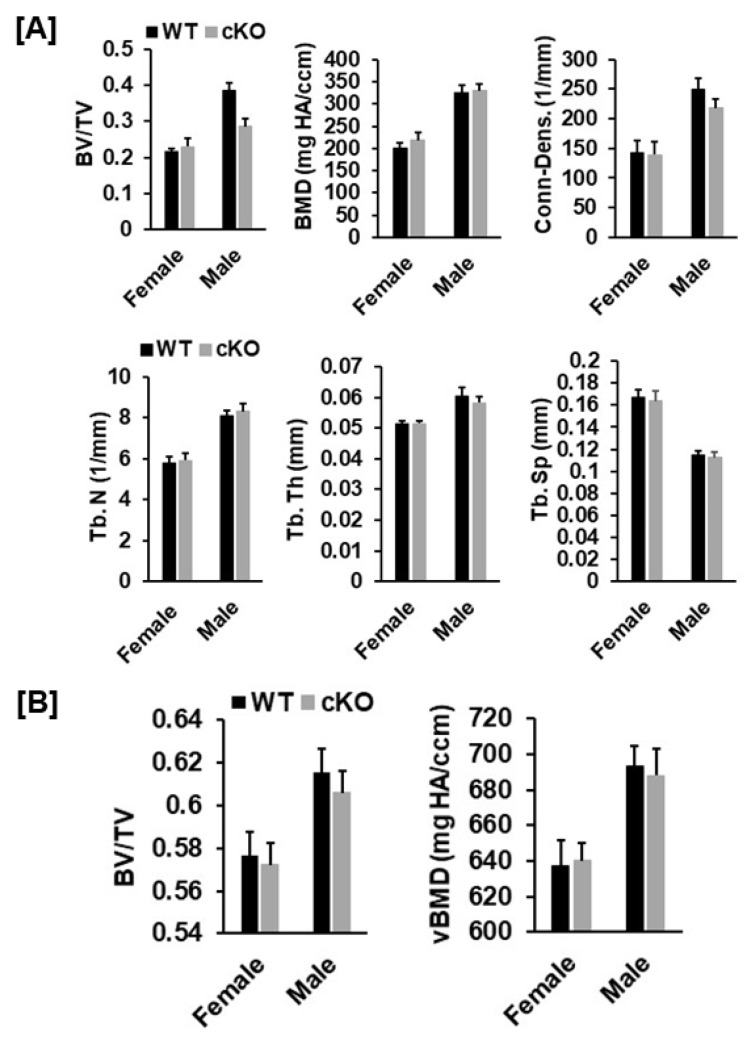
Micro-CT analysis revealed no changes in trabecular and cortical parameters of the tibia of the *Phd3* cKO mice at 16 weeks of age. (**A**): Quantitative data of trabecular parameters of the tibia (BV/TV, BMD, Conn-Dens., Tb. N, Tb. Th, and Tb. Sp). (**B**): Quantitative data of cortical BV/TV and BMD of the tibia.

**Figure 6 cells-10-02200-f006:**
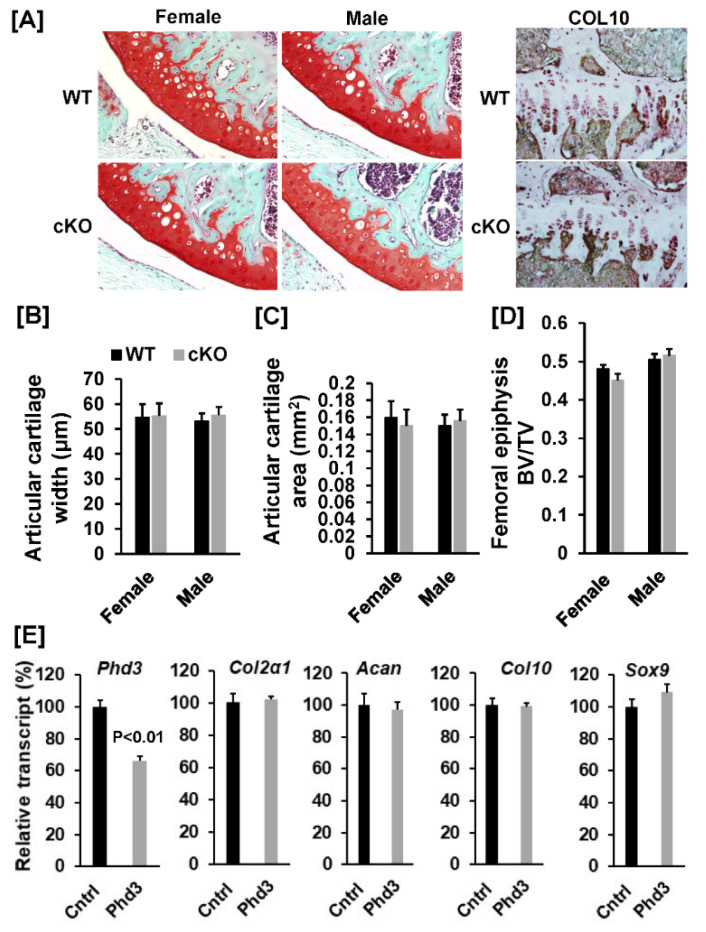
Knockdown of *Phd3* expression did not influence chondrocyte differentiation. (**A**): Bone sections from the distal femurs of the WT and cKO female mice were stained with safranin red and methyl green. Articular cartilage was stained as red. Bone sections from the distal femurs of the WT and cKO female mice were also immuno-stained with Collagen 10 (Col10) (primary antibody dilution 1:100). Positive Col10-positive chondrocytes in the growth plates were stained as red. (**B**,**C**): Quantitative data of articular cartilage width and area of the distal femur. (**D**): Micro-CT data of the femoral epiphysis. (**E**): Knockdown of *Phd3* expression in chondrocytes did not affect expression of *Col2α1*, *Acan,*
*Col10α1* or *Sox9*. Primary chondrocytes were transduced with Lentivirus-shRNA against *Phd3* or non-specific control Lentivirus shRNA. The transduced cells were differentiated for 3 days, followed by RNA extraction for real time PCR. Data normalized to chondrocytes transduced with control lentiviral shRNA particles. A = *p* < 0.05 vs. control shRNA (*n* = 4).

**Table 1 cells-10-02200-t001:** Primer sequences used for real time PCR.

Gene	Forward Primer	Reverse Primer
*Ppia*	5′-CCATGGCAAATGCTGGACCA	5′-TCCTGGACCCAAAACGCTCC
*Phd3* *Phd2*	5′-GGGACGCCAAGTTACACGGA 5′-GAAGCTGGGCAACTACAGGA	5′-GGGCTCCACGTCTGCTACAA 5′-CATGTCACGCATCTTCCATC
*Col2*	5′-TGGCTTCCACTTCAGCTATG	5′-AGGTAGGCGATGCTGTTCTT
*Col10*	5′-ACGGCACGCCTACGATGT	5′-CCATGATTGCACTCCCTGAA
*Acan*	5′-GACCAGGAAGGGAGGAGTAG	5′-CAGCCGAGAAATGACACC
*Sox9*	5′-CGGAGGAAGTCGGTGAAGA	5′-GTCGGTTTTGGGAGTGGTG

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
