# Peer review of "Prolyl Hydroxylase Domain-Containing Protein 3 Gene Expression in Chondrocytes Is Not Essential for Bone Development in Mice"

_cells, 2021, doi:10.3390/cells10092200_

Round 1

Reviewer 1 Report

The authors previously detected an upregulation of Phd3 in Phd2 knockout chondrocytes that showed an increase in mineralization. In the current study, to test whether Phd3 is involved in endochondral bone formation, the authors describe a mouse where Phd3 is conditionally knocked out in chondrocytes. The authors do not find any significant skeletal changes in mice with Phd3 knockout chondrocytes and littermate controls. This study adds some clarification to the role of Phds in endochondral ossification.

The manuscript is well written, aside from a few small grammatical errors that can be easily fixed with another read through by the authors. Some data also seems to be missing.

Main Comments:

  1. Pg 2, line 66 – do you mean Col1a2-Cre or Col2a1-Cre in this sentence?
  2. Add the product description for anti-PHD3 antibody in methods section.
  3. In figure 2), reference is made to “[D] Expression PHD3 in chondrocytes of the distal femoral growth plates of the WT and cKO, detected by immunohistochemistry.” I don’t see a part [D] in the figure.
  4. Pg 11, line 249 – “Can” should be “Acan”; line 251 - “RAN extraction”, should say “RNA extraction”. Check grammar throughout article.
  5. What is the effect of Phd3 (or other Phds) over-expression in chondrocytes?

Author Response

We thank the reviewer for her/his positive remarks.

  1. Pg 2, line 66 – do you mean Col1a2-Cre or Col2a1-Cre in this sentence?

Response: we apologize for the error. It is Col2a1-Cre that is targeted in chondrocytes. We have corrected the error in the revised manuscript.

2. Add the product description for anti-PHD3 antibody in methods section.

Response: We apologize for the mistake. We applied immunohistochemistry to examine chondrocyte marker gene expression of collagen 10. We have corrected the error in the revised manuscript. Anti-PHD3 antibody (Catalog Number: MAB6954 from R&D Systems) that we used did not detect PHD3 expression in the paraffin sections of mouse bone and, therefore, we did not include this result in the manuscript.

3. In figure 2), reference is made to “[D] Expression PHD3 in chondrocytes of the distal femoral growth plates of the WT and cKO, detected by immunohistochemistry.” I don’t see a part [D] in the figure.

Response: We apologize for the error.  As stated above, we were unsuccessful in detecting PHD3 expression in the paraffin sections of mouse bone using commercial antibody preparations against PHD3.  We, therefore, did not include this result.  We have now deleted this statement in the revised manuscript.

4. Pg 11, line 249 – “Can” should be “Acan”; line 251 - “RAN extraction”, should say “RNA extraction”. Check grammar throughout article.

Response: We thank the review for pointing out the errors. We have now corrected the errors in the revised manuscript.

5. What is the effect of Phd3 (or other Phds) over-expression in chondrocytes?

Response: We did not test the effect of  PHD overexpression in chondrocytes in vitro and in vivo. However, we previously observed elevated expression of PHD3 in the Col2a1-Cre mediated Phd2 cKO articular cartilage and in growth plate chondrocytes both in vivo and in vitro. In this study, we conditionally knocked out Phd3 gene in chondrocyte lineage cells to examine the role of Phd3 and found the knockout of Phd3 gene in chondrocyte had no effect on chondrocyte differentiation and bone development. We will confirm the role of PHD2 deficiency-induced Phd3 overexpression in chondrocyte differentiation and growth plate development in Phd2/3 double knockout mice in future.

Reviewer 2 Report

The study deals with the role of chondrocytic prolyl hydroxylase domain-containing protein 3 (Phd3) expression in bone development in mice. To determine if the Phd3 expression plays a role in endochondral bone formation, the authors deleted the Phd3 gene in chondrocytes crossing Phd3 floxed mice with Col2a1-Cre mice. The authors revealed by microCT and histology analyses of the bone phenotype of Cre+;Phd3/f/f mice that Phd3 expression is not essential for the regulation of bone development in mice. The authors demonstrated previously that conditional disruption of the Phd2 gene in the chondrocyte lineage led to an increase in long bone trabecular bone mass and loss of Phd2 expression presumably resulted in a compensatory increase of Phd3 expression. The authors conclude from the latter observation that the increased expression of Phd3 could be at least partly involved in the development of the trabecular phenotype of Cre+;Phd2/f/f mice. Therefore, the study is of great importance.

Minor notes

Is the expression of Phd2 affected in the mice with Phd3 deficieny in chondrocytes?

Fig3E: Could the unchanged expression of the chondrocytic differentiation markers be influenced by the low efficiency of the Phd3 knockdown in the primary chondrocytes? The authors should discuss the expression results.

Author Response

We thank the reviewer for her/his positive remarks. 

  1. Is the expression of Phd2 affected in the mice with Phd3 deficiency in chondrocytes?

Response: We thank the reviewer for her/his suggestion. We have now added the data in figure 1B, showing the expression level of Phd2 was not affected in the epiphyses of Phd3 deficient mice as compared to the WT control mice.

  1. Fig3E: Could the unchanged expression of the chondrocytic differentiation markers be influenced by the low efficiency of the Phd3 knockdown in the primary chondrocytes? The authors should discuss the expression results.

Response: We thank the reviewer for her/his suggestion. We have acknowledged the partial knock down of Phd3 expression in chondrocytes in the revised manuscript. We stated “in future, we will confirm the role of PHD2 deficiency-induced Phd3 overexpression in chondrocyte differentiation and growth plate development in Phd2/3 double knockout mice. We will also verify the expression levels of the chondrocyte differentiation markers in primary chondrocytes derived from the Phd3 global KO mice.